



# Daily variation in net primary production and net calcification in coral reef communities exposed to elevated pCO₂

[1,2] Steve Comeau, [1] Peter J. Edmunds, [1,3] Coulson A. Lantz, [1] Robert C. Carpenter

[1] Department of Biology, California State University, 18111 Nordhoff Street, Northridge,

CA 91330-8303, USA.

[2] School of Earth Sciences, Oceans Institute and ARC Centre of Excellence for Coral

Reef Studies, The University of Western Australia, Crawley, Western Australia 6009,

Australia

[3] Southern Cross University School of Environment, Science, and Engineering, Military

Road Lismore NSW 2480 Australia

Correspondence to: Steve Comeau (steve.comeau@uwa.edu.au)



**Abstract**

The threat represented by ocean acidification (OA) for coral reef has received
considerable attention because of the sensitivity of calcifiers to changing water carbonate
chemistry. However most studies have focused on the organismic response of

calcification to OA, and only a few have addressed community-level effects, or
investigated parameters other than calcification, such as photosynthesis. Light
(Photosynthetically Active Radiation, PAR) is a driver of biological processes on coral
reefs, and the possibility that these processes might be perturbed by OA has important
implications for community function. Here we investigate how $CO_2$ enrichment affects

the relationships between PAR and community net $O_2$ production ($P_{net}$), and between
PAR and community net calcification ($G_{net}$), using experiments on three coral
communities constructed to match (i) the back reef of Moorea, French Polynesia, (ii) the
fore reef of Moorea, and (iii) the reef flat of Oahu, Hawaii. The results were used to test
the hypothesis that OA affects the relationship between $P_{net}$ and $G_{net}$. For the three

communities tested, $pCO_2$ did not affect the $P_{net}$-PAR relationship, but it affected the
intercept of the hyperbolic tangent curve fitting the $G_{net}$-PAR relationship for both reef
communities in Moorea (but not in Oahu). For the three communities, the slopes of the
linear relationships between $P_{net}$ and $G_{net}$ were not affected by OA, although the
intercepts were depressed by the inhibitory effect of high $pCO_2$ on $G_{net}$. Our result

indicates that OA can modify the balance between net calcification and net
photosynthesis of reef communities by depressing community calcification, but without
affecting community photosynthesis.



## 1. Introduction

Ocean acidification (OA), which is caused by the dissolution of atmospheric $CO_2$

in surface seawater, leads to profound changes in seawater carbonate chemistry,

involving an increased concentration of bicarbonate ions and dissolved $CO_2$, and a

decrease in concentration of carbonate ions and pH (Feely et al. 2004). The effects of

these changes on tropical coral reefs are beginning to be understood in detail, with most

studies reporting a decrease in calcification of scleractinian corals and coralline algae at

reduced seawater pH (Gattuso and Hanson 2011; Kroeker et al. 2013).

         To date, studies addressing the effects of OA on coral reefs have been performed

mostly at the scale of individual organism, and have focused on calcification as a

response variable (Schoepf et al. 2013; Comeau et al. 2013; Okazaki et al. 2016), while

studies focusing on larger spatial scales (i.e., whole communities) have remained rare,

mostly because of technical constraints (e.g., Dove et al. 2013; Comeau et al. 2015,

2016a). The few experiments addressing the effects of OA on intact coral reef

communities have confirmed the threat to calcification rates previously reported for

individual organisms, notably by showing a decreased capacity of communities to

maintain positive net calcification under conditions mimicking future ocean in which

seawater pH will be depressed 0.15 – 0.3 units relative to present-day conditions (e.g.,

Dove et al. 2013; Comeau et al. 2015, 2016a). These community-level studies have

focused mostly on the response of calcification to low pH (Dove et al. 2013; Comeau et

al. 2015, 2016a) and, in contrast, the effect of increasing $pCO_2$ on community net $O_2$



production has rarely been investigated. Where this issue has been addressed, community

$O_2$ production has been found to be insensitive to $pCO_2$ (to ~ 1000 µatm) (Leclerc et al.

2002; Langdon and Atkinson 2005, Dove et al. 2013), while a positive effect of $pCO_2$ on

the net production of photosynthetically fixed organic carbon has been reported during a

flume experiment (Langdon and Atkinson 2005).

        Investigating the combined response to OA of primary production and

calcification of benthic coral reef communities is critical, because increasing dissolved

$CO_2$ and bicarbonate ion concentrations potentially could "fertilize" photosynthesis of

marine organisms (Connell and Russell 2010; Hepburn et al. 2011; Connell et al. 2013),

thereby perturbing ecosystem trophodynamics. A stimulatory effect of OA on

photosynthesis could, for calcifying taxa such as corals and coralline algae, support

higher rates of calcification by increasing the ease with which the metabolic costs of

these events could be met through enhanced respiration fuelled by greater availability of

carbon substrates (Comeau and Cornwall 2016). However, a stimulatory effect of OA on

photosynthesis have not been clearly established for coral reef organisms, and to date, the

evidence in support of this possibility is equivocal (e.g., Anthony et al. 2008; Kroeker et

al. 2013; Comeau et al. 2016b).

One reason why studies of the effect of $pCO_2$ on the relationship between primary

production and calcification are technically challenging is that the relationships between

light (Photosynthetically Active Radiation, PAR) and both photosynthesis and

calcification are non-linear (e.g., Borowitzka 1981; Chalker et al. 1988; Muscatine 1990;



Chisholm 2000). In symbiotic reef corals, the relationships between photosynthesis and

PAR, and between calcification and PAR, generally are best fit by a hyperbolic tangent

function (Chalker 1981; Marubini et al. 2001), which is characterized by a rapid rise of

photosynthesis (or calcification) with initial increases in PAR from darkness, followed by

a plateau of response at saturating light, and sometimes a reduction in response at the

highest PAR intensity (i.e., photoinhibition [e.g., Brown et al. 1999]). No studies have

investigated the effect of $pCO_2$ enrichment on the mathematical parameters defining the

hyperbolic tangent relationship between PAR and photosynthesis (or calcification) for

coral reef organisms and communities.

Because calcification of coral reef communities is coupled to photosynthesis on

timescales of hours-to-days (Gattuso et al. 1999), examination of high frequency

variation in the net $O_2$ production ($P_{net}$)- net calcification ($G_{net}$) relationships for these

communities has the potential to reveal the capacity to respond dynamically to varying

conditions (i.e., Jokiel et al. 2014). The relationship between $P_{net}$ and $G_{net}$ for coral reefs is

relatively well known at the community level, and generally describes a positive linear

relationship (Gattuso et al. 1999; Falter et al. 2012). Such a relationship reflects emergent

properties arising from the stimulation of $G_{net}$ by $P_{net}$ at the organism scale (i.e., for corals

and calcified algae) (Jokiel et al. 2014), most likely because $P_{net}$ can supply the carbon

resources necessary as substrates for aerobic respiration (Stambler 2011), modify the

intracellular and surrounding seawater chemistry (Marubini et al. 2008; Jokiel et al.

2014), and provide the building blocks necessary to construct the organic matrix found

within coral skeletons (Muscatine et al. 2005). Unfortunately, it is difficult to test the



hypothesis that the $G_{net}$ - $P_{net}$ relationship for reef communities is affected by carbonate

chemistry, because the seawater chemistry varies with $P_{net}$ in the natural environment

(Jokiel et al. 2014; Shaw et al. 2015). To test for an effect of seawater carbonate

chemistry on the $G_{net}$ - $P_{net}$ relationhip of reef communities, it is therefore necessary to

first conduct experiments in a controlled environment to assess how seawater carbonate

chemistry alone affects the $G_{net}$ - $P_{net}$ relationship.

        The present study tests the hypothesis that the enrichment in seawater $pCO_2$ due

to OA will affect the relationships between $P_{net}$ and *PAR,* and between $G_{net}$ and *PAR* for

intact reef communities fabricated in outdoor flumes (sensu Atkinson et al. 1994). The

second hypothesis tested is that the $P_{net}$ - $G_{net}$ relationships would be affected by OA,

based on the rationale that community $P_{net}$ and $G_{net}$ would respond in dissimilar ways to

high $pCO_2$. Because the shape of these relationships likely depends on the community

composition (i.e., the taxa present and their relative abundances [Gattuso et al. 1999]), we

used results from three independent experiments to explore variations in the relationships

caused by differences in environmental conditions and differences in the taxonomic

assemblages composing the communities tested.  Data from three experiments conducted

in flumes in two locations in the tropical Pacific were combined; one experiment focused

on a back reef community assembled in Moorea, French Polynesia, during the Austral

spring 2013 (Comeau et al. 2015); one experiment focused on a reef flat community

assembled in Kaneohe Bay, Oahu, during the winter 2014; and one experiment focused

on a fore reef community assembled in Moorea, during the Austral spring 2014 (Comeau

et al. 2016a). For the communities analysed in Moorea, the present contribution describes



in more detailed the results for net calcification, as well as new results for photosynthesis,

that originate from experiments that are described in part in previous papers (Comeau et

al. 2015, 2016a); the study conducted in Oahu has not been described before. The three

communities were incubated in outdoor flumes of similar designs, and were operated

under ambient and elevated $pCO_2$ (~ 400 µatm and ~1300 µatm, respectively).  When the

experiments were conducted, community $P_{net}$ and $G_{net}$ were measured simultaneously.

## 2. Materials and Methods

### 2.1 Collection and sample preparation


        This study utilizes results from three experiments conducted between August

2013 and October 2014. The first and third experiments were carried out in Moorea,

French Polynesia, at the Richard B. Gump South Pacific Research Station, and the second

experiment was conducted in Oahu, Hawaii, on Coconut Island at the Hawaii Institute of

Marine Biology.

        The first experiment took place in August-October 2013, and focused on a back

reef community from 1–2 m depth on the north shore of Moorea (Comeau et al. 2015).

This community consisted of massive *Porites* spp. (11% cover), *Porites rus* (6%),

*Montipora* spp. (3%), *Pocillopora* spp. (2%), and crustose coralline algae (6%), and the

coverage of each taxon in the flume was scaled to represent the community structure

measured in this habitat in 2013 (Carpenter 2014; Edmunds 2014). To create benthic

communities that were ecologically relevant to the back reef of Moorea (Carpenter 2014;



Edmunds 2014), two-third of the area of the working section of the flume was occupied

by sediments collected from the lagoon at 2-m depth using custom made sediment boxes

($0.4 \times 0.3$ m in area and 0.3 m deep).

The second experiment was carried out in Oahu in January-February 2014 and

focused on a benthic community similar to that found at 1-2 m depth on the Kaneohe Bay

barrier reef flat in 2013. This community consisted of *Porites compressa* (7% cover),

*Montipora capitata* (12%), massive *Porites* spp. (3%), and *Pocillopora damicornis* (2%),

and the crustose coralline alga *Porolithon onkodes* (4%) (Jokiel et al. 2015). As

described above for experiment 1, sediments were inserted into the floor of the flume to

recreate ecologically relevant communities. Since the flumes in Oahu (as designed and

utilized by M Atkinson (e.g., Atkinson et al. 1994)) were not designed to include

sediments, a custom-made sediment box was inserted into the floor of the flumes to

provide an area occupying two-thirds of the floor of the working section of the flume

with sediment to a depth of ~ 5-8 cm.

The third experiment was carried out from August to October, 2014 in Moorea,

and focused on outer reef benthic communities prepared from specimens collected from ~

15–17-m depth (Comeau et al. 2016a). This community consisted of *Pocillopora* spp.

(11% cover), massive *Porites* spp. (8%), and *Acropora* spp. (8%), and the crustose

coralline alga *Porolithon onkodes* (5%), and the coverage of each taxon was scaled in the

flumes to match community structure recorded in this habitat (at the same depth) in 2006

(Carpenter 2014; Edmunds 2014). Because the benthos on the fore reef of Moorea





consisted mostly of reef pavement in 2006 (i.e., cemented calcium carbonate substratum

covered by algal turf and coralline algae (Carpenter 2014; Edmunds 2014), 55% of the

floor of the flume was covered by ~ 20 × 20 × 5 cm pieces of reef pavement collected

from ~15-m depth to match the % cover of the benthic community.  Sediments were not

included in this experiment, as they are not common on the fore reef at the investigated

depth.

In Moorea, the two experiments were performed in four outdoor flumes consisting

of a working section of 5.0 × 0.3 × 0.3 m (as in Comeau et al. 2015) in which water was

re-circulated at a constant speed of $10 \pm 0.5$ cm s$^{-1}$ (mean ± SE; Experiment 1) or $8 \pm 0.5$

cm s$^{-1}$ (Experiment 3) that represented the mean in situ flow speed over the year

measured in the two habitats (Washburn 2014; Comeau et al. 2016). Two flumes were

maintained at ambient $pCO_2$ (~ 400 µatm), and two at elevated $pCO_2$ (~1200–1300 µatm,

see below). Fresh sand-filtered seawater (nominal pore size of ~ 100 µm) was dispensed

continuously into the flumes at 5 L min$^{-1}$, and the experiments lasted eight (Experiment

1) or seven weeks (Experiments 3).

In Oahu, the benthic community was constructed in two outdoor flumes, one with

a working section of 9 × 0.6 × 0.3 m, and one with a working section of 4 × 0.4 × 0.4 m;

one of these flumes was maintained at ambient $pCO_2$ and one at elevated $pCO_2$. To

address the confounding effect of flumes in this design (i.e., the flumes were allocated to

one of two treatments and the flumes were not of an identical design), the first

experiment ended after three weeks, the $pCO_2$ treatments were switched between flumes,

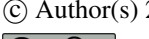



and new communities (with the same taxon composition including sediment) were placed

in the two flumes for a second trial of the same experiment lasting 3 weeks. Fresh sand-

filtered seawater was dispensed continuously into both flumes (at 5-10 L min$^{-1}$), and a

flow speed of 10 cm s$^{-1}$, similar to that employed in the earlier trial with the back reef

communities of Moorea, was maintained using electric trolling motors (Minnkota Riptide

55, Minnkota, USA).

The three experiments were performed outdoors under natural sunlight that was

attenuated using shade cloth to maintain PAR values similar to ambient PAR recorded in

situ in each habitat. In Experiment 1 and 2, the maximum PAR was set at ~ 1000 μmol

quanta m$^{-2}$ s$^{-1}$ to represent light levels at ~ 1–2m depth in the back reef (Carpenter et al.

2016), and in Experiment 3, maximum PAR was set at ~ 600 μmol quanta m$^{-2}$ s$^{-1}$ to

mimic light levels recorded at 17-m depth on the fore reef of Moorea around noon on a

cloudless day (Carpenter et al. 2016). For Experiment 3 (with an outer reef community

from deeper water), blue acetate filters (Lee Filters #183 Moonlight Blue) were placed

over the flumes to filter ambient sunlight in the 600-800 nm range to approximate the

light spectrum found at 17-m depth (Comeau et al. 2016a). Temperature in all flumes was

maintained at ambient seawater temperature when the experiments were conducted,

which corresponded to ~ 27 °C in Experiment 1 and 3 (both conducted during Austral

spring) and ~ 24 °C in Experiment 2 (conducted in winter).


*2.2 Carbonate chemistry manipulations and measurements*



For the three experiments, $pCO_2$ levels were chosen to match ambient $pCO_2$ (~

400 µatm) and the $pCO_2$ expected in the atmosphere by the end of the present century

under a pessimistic scenario of further anthropogenic activity (Representative

Concentration Pathway 8.5, ~1300 µatm, Moss et al., 2010). $pCO_2$ in the flumes was

controlled using pH controllers (Aquacontroller, Neptune systems, USA) that controlled

the delivery of either pure $CO_2$ or $CO_2$-free air into the seawater. To match the natural

diel variation in seawater pH in shallow back reef communities (Hofmann et al., 2011;

Comeau et al., 2014a), in Experiment 1 and 2, seawater pH was maintained 0.1 unit lower

at night (from 18:00 to 6:00) than during the day. Diel variation in pH was not applied

during Experiment 3, because seawater pH varies < 0.1 between day and night on the fore

reef of Moorea (S. Comeau unpublished data).

For the three experiments, pH on the total scale ($pH_T$) was measured daily using a

portable pH meter (Orion 3-stars, Thermo-Scientific, USA) fitted with a DG 115-SC pH

probe (Mettler Toledo, Switzerland) calibrated every other day with Tris/HCl buffers

(Dickson et al., 2007). $pH_T$ also was measured every 2 weeks spectrophotometrically

using m-cresol dye (Dickson et al., 2007). Mean values of $pH_T$ measured

spectrophotometrically and using a pH electrode differed < 0.02 pH units. Total alkalinity

($A_T$) was measured using open-cell potentiometric titrations (Dickson et al., 2007) on 50-

mL samples of seawater collected every 2-3 d. Accuracy of $A_T$ measurements was

checked by titrating certified reference materials provided by A. G. Dickson (batch 122

and 140) that yielded $A_T$ values within ~ 4 µmol kg$^{-1}$ of the nominal value. Parameters of



the carbonate system in seawater were determined with the R package seacarb (Gattuso et

al., 2015) using measured values of $pH_T$, $A_T$, temperature, and salinity.

*2.3 Net calcification and primary production measurements*

Net community calcification ($G_{net}$) in the flumes was measured using the total

alkalinity anomaly method (Chisholm and Gattuso 1991; Schoepf et al. 2016), and net

community primary production ($P_{net}$) was measured using oxygen sensors (TROLL 9500,

In-Situ) that measured the $O_2$ concentration at 60-second intervals. Measurements of

changes in dissolved inorganic carbon (DIC) were not meaningful with our experimental-

design because DIC was held constant by adding pure $CO_2$ during the incubations to

maintain $pCO_2$ at target values.

For the three experiments, community metabolism was measured every 7 d using

single 24-h incubations during which the addition of seawater to the flumes was stopped,

and the flumes were operated in a closed circuit mode. During these incubations,

seawater samples for the determination of $A_T$ were taken every 3 h during the day, and

every 6 h at night, to estimate $G_{net}$, while $O_2$ was constantly monitored. Since only two $O_2$

sensors were available, and experiments were conducted in four flumes in Moorea, $P_{net}$

was measured for each incubation in one ambient and one elevated $pCO_2$ flumes that

were picked randomly. In Oahu, one $O_2$ sensor was used in each flume during the

incubations. Acrylic covers placed on top of the flumes limited gas exchange with the

atmosphere but did not prevent it. However, gas exchange, which was similar between

treatments, was estimated (using the methods of Langdon and Atkinson [2005]) to be





minimal (i.e. < 5%) and, therefore, was not taken into account in the present study. To

maintain $A_T$, nutrient concentrations, and $pO_2$ at values close to ambient seawater in the

sampled habitats, ~ 50% of the flume volume was replaced every 3 h during the day, and

every 6 h at night (i.e., at 6:00, 9:00, 12:00, 15:00, 18:00, and 00:00). Light was

monitored constantly during the incubations using cosine-corrected PAR sensors

(Odyssey, Dataflow Systems Pty Ltd, Christchurch, New Zealand).

*2.4 Calculations and statistical analysis*

$P_{net}$ was estimated hourly by calculating the change in $O_2$ during the incubations,

except for the hours during which the seawater was refreshed (6:00, 9:00, 12:00, 15:00,

18:00, and 00:00 hrs). $G_{net}$ was estimated at 3 h intervals during the day and 6 h intervals

at night by collecting $A_T$ samples at the beginning (after seawater refreshing) and at the

end of each incubation (before adding fresh seawater).

Because there were no significant differences in calcification between flumes for

each treatment (Comeau et al. 2015, 2016a), $G_{net}$ was pooled among replicate flumes in

each treatment. $P_{net}$ was measured in Moorea in only one flume per treatment at a time,

and it was assumed that the measurements represented the average response to the

conditions experienced in each treatment. Individual measurements of $G_{net}$ and $P_{net}$ in

Oahu were considered replicates.

An Akaike Information Criterion (AIC) approach was used to determine if a

linear, logarithmic, or hyperbolic tangent functions best described the functional

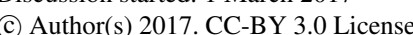



relationships between $P_{net}$ and PAR, and between $G_{net}$ and PAR, for each community (see

details in Comeau et al. 2013). A linear relationship was fit to explore a "proportional

effect" model for increasing PAR. A logarithmic function and a hyperbolic tangent

function that are commonly used to describe the relationship between $P_{net}$ and PAR for

reef corals (Chalker 1981; Marubini et al. 2001), also were fit to the data in cases where

photosynthesis (or calcification) initially rapidly increased with PAR, then approached an

asymptote at saturating PAR.

The hyperbolic tangent function between PAR and $P_{net}$ in the light corresponded

to:

$$P_{net} \; = \; C_0 \; + P_{net\,max}\tanh\frac{(\alpha\,I)}{P_{net\,max}}$$

where $P_{net\,max}$ is the maximum photosynthetic rate, $I$ is the PAR, $\alpha$ is the slope of the

initial portion of the $P_{net}$ versus $I$ relationship, and $C_0$ is the intercept.

Similarly, the hyperbolic tangent function for the relationship between PAR and

$G_{net}$ in the light was:

$$G_{net} \; = \; C_0 \; + G_{net\,max}\tanh\frac{(\alpha\,I)}{G_{net\,max}}$$

where $G_{net\,max}$ is the maximum calcification rate, $I$ is the PAR, $\alpha$ is the slope of the initial

portion of the $G_{net}$ versus $I$ relationship, and $C_0$ is the intercept.

The best fits of the functions (least squares) were determined using the function

*nls* in R, and t-tests were used to compare the curve parameters between $pCO_2$ treatments.




To test the hypothesis that $P_{net}$ and $G_{net}$ were associated, mean $P_{net}$ corresponding to the $G_{net}$ determination intervals (3 h periods during the day and 6 h at night) were calculated, and the relationship between $P_{net}$ and $G_{net}$ was investigated using a correlation approach (sensu Gattuso et al. 1999). When the linear associations between $G_{net}$ on $P_{net}$ were significant, analyses of covariance (ANCOVA), with $P_{net}$ as the covariate, were used to test the effects of $pCO_2$ (a fixed effect) on the $P_{net}$ - $G_{net}$ relationship for each experiment. All analyses were performed using R software (R Foundation for Statistical Computing). In this design, both $P_{net}$ and $G_{net}$ are random variables for which a test of association is best accomplished with correlation. Evaluating the slope and intercept is problematic as it is not appropriate to use Model I (least squares) approaches for the purpose of describing the functional relationship between two random variables. In the present case, we report Model I slopes because we are interested in the capacity to predict $G_{net}$ from $P_{net}$ and because Model I slopes are integral to the ANCOVA approach.

## 3. Results

Carbonate chemistry was tightly controlled during the three experiments, with mean $pCO_2$ maintained at 453 ± 30, 460 ± 23, and 400 ± 14 µatm in the ambient treatments, and 1317 ± 50, 1233 ± 76, and 1176 ± 37 µatm in the elevated $pCO_2$ treatments during Experiments 1, 2, and 3, respectively (all ± SE, n = 42–56). In all experiments and both treatments, aragonite saturation states ($\Omega_{arag}$) was ~ 3.52, 2.59, and 3.71 in the ambient treatments, and 1.64, 1.36, and 1.75 in the elevated $pCO_2$ treatments during Experiments 1, 2, and 3, respectively (Table 1). $\Omega_{arag}$ was lower during



Experiment 2 in Oahu compared to Experiments 1 and 3 in Moorea because of naturally

lower $A_T$ (~ 2160 µmol kg$^{-1}$) and temperature (~24°C) in this location (cf in Moorea

where $A_T$ is ~2340 µmol kg$^{-1}$ at 27°C).

For the three experiments, benthic community structure was not measured during

these short experiments, but we assume that changes were minor as there was no coral

mortality and planar growth would have been trivial over several weeks.

*3.1 Relationships of $P_{net}$ and $G_{net}$ with PAR*

AIC analyses justified the use of a hyperbolic tangent function (versus linear or

logarithmic functions) to fit the relationship between $P_{net}$ and PAR during the day for the

three experiments under the two pCO$_2$ conditions (Fig. 1A, B, and C). For the back reef

community of Moorea, the back reef community of Oahu, and the fore reef community of

Moorea, there was no effect of pCO$_2$ on any of the parameters of the relationship between

$P_{net}$ and PAR (Table 2).


AIC tests also confirmed that the relationships of $G_{net}$ with PAR could be fit with

a hyperbolic tangent function for the three experiments under the two pCO$_2$ conditions

tested (Fig. 2A, B, and C). For the Moorea back reef community, there was no difference

in maximum calcification ($G_{net\ max}$), and slope of the initial portion of the relationship (α)

between pCO$_2$ treatments (Table 2). However, pCO$_2$ affected the intercepts ($C_0$, p = 0.

046), with $C_0$ at ambient pCO$_2$ (1.26 mmol m$^{-2}$ h$^{-1}$) greater than $C_0$ at elevated pCO$_2$ (-





0.52 mmol m$^{-2}$ h$^{-1}$). The relationship of $G_{net}$ with PAR for the back reef communities in Oahu was not statistically affected by pCO$_2$ (Table 2). For the fore reef community of Moorea, $G_{net\ max}$ and α did not differ between treatments, but $C_0$ was higher (2.77 mmol

O$_2$ m$^{-2}$ h$^{-1}$) at ambient versus elevated pCO$_2$ (0.58 mmol O$_2$ m$^{-2}$ h$^{-1}$) (Table 2).

*3.2 Relationships between $P_{net}$ and $G_{net}$*

   For the back reef communities of Moorea, the relationship between $P_{net}$ and $G_{net}$ were significantly and positively correlated (p < 0.001 under ambient and elevated pCO$_2$)

with slopes of 0.17 ± 0.03 mmol CaCO$_3$ mmol O$_2$$^{-1}$ under ambient pCO$_2$, and 0.18 ± 0.03 mmol CaCO$_3$ mmol O$_2$$^{-1}$ (both ± SE, n = 48) under elevated pCO$_2$ (Fig. 3A). There was no difference between treatments in slopes (ANCOVA, p = 0.749, Table 3), but elevations were 61% greater under ambient versus elevated pCO$_2$ (p < 0.001, Table 3).

$G_{net}$ and $P_{net}$ for the back reef communities of Oahu also were positively correlated (p < 0.001 under both ambient and elevated pCO$_2$) and their relationships exhibited slopes of 0.14 ± 0.02 mmol CaCO$_3$ mmol O$_2$$^{-1}$ under ambient pCO$_2$, and 0.17 ± 0.02 mmol CaCO$_3$ mmol O$_2$$^{-1}$ (both ± SE, n = 36) under elevated pCO$_2$ (Fig. 3B). There was no difference between treatments in slopes (ANCOVA, p = 0.286, Table 3), but the

elevations were 32% greater under ambient versus elevated pCO$_2$ (p < 0.001).

   For the fore reef community of Moorea, the relationships between $G_{net}$ and $P_{net}$ were significant under ambient and elevated pCO$_2$ (p < 0.001) and had respective slopes of 0.27 ± 0.05 mmol CaCO$_3$ mmol O$_2$$^{-1}$, and 0.30 ± 0.06 mmol CaCO$_3$ mmol O$_2$$^{-1}$ (both ±



SE, n = 28). For the back reef communities, there were no difference of the slopes

between $G_{net}$ and $P_{net}$ between treatments (ANCOVA, p = 0.623, Table 3), but elevations

were 48% greater under elevated versus ambient $pCO_2$ (p = 0.002).

## 4. Discussion


By testing the response of three coral reef communities to OA under natural PAR,

our study demonstrates that the relationships between $P_{net}$ and PAR and $G_{net}$ and PAR for

back reef and outer reef communities are not affected by $pCO_2$. Our results also

demonstrate that the slope of the relationship between $P_{net}$ and $G_{net}$ was unaffected by

increasing $pCO_2$, in contrast the intercepts were more elevated in the ambient treatments.

Such results were caused by a constant effect of OA on $G_{net}$ for the range of $P_{net}$ values

measured in the three communities.

For the three assembled communities tested, $pCO_2$ did not affect the functional

relationship between PAR and $P_{net}$ as modelled using a hyperbolic tangent function. This

result suggests that for the organisms composing the three communities investigated, the

additional quantities of bicarbonate and dissolved $CO_2$ available under OA conditions did

not enhance photosynthesis across the range of light intensities and community structures

tested. However, as our results come from experiments completed in a single season, we

cannot be sure whether the results are consistent throughout the year, as seasonal

variations in community and organismic $P_{net}$ and $G_{net}$ are common on coral reefs (e.g.,

Falter et al. 2012). Whether increasing $pCO_2$ has beneficial consequences for rates of



photosynthesis of marine organisms is equivocal (Connell and Russell 2010; Britton et al.

2016) and, indeed, the absence of an effect of $pCO_2$ on photosynthesis may have

important biological meaning (e.g., Kroeker et al. 2013). For instance, such an outcome

could reflect the presence of diverse carbon concentrating mechanisms (CCM), which

allow organisms to actively concentrate $CO_2$ at the site of Rubisco activity by actively

transporting $HCO_3^-$ across internal membranes (Giardano et al. 2005; Raven et al. 2014).

Increases in concentration of dissolved $CO_2$ in seawater that occur as a result of OA

(Feely et al. 2004) could have beneficial consequences for photosynthetic rates of species

that currently are DIC limited (Diaz-Pulido et al. 2016), because these organisms often

rely on inefficient and energetically costly CCMs to access $CO_2$ (Raven et al. 2014).

      The present study, as well as previous studies of both coral reef organisms (corals

and calcified algae) (Schneider & Erez 2006; Comeau et al. 2016b), and coral reef

communities (Leclercq et al. 2002; Langdon et al 2003; Dove et al. 2013), showed no

change in $P_{net}$, measured by $O_2$ changes, in response to OA arising from $pCO_2$ values as

high as 2000 µatm. Stimulatory effects of $pCO_2$ on $P_{net}$ probably were not detected in our

communities (i.e., where coral cover ranged form 22–27%), because such effects are

likely to be minimal for endosymbiotic *Symbiodinium* in corals that possess a CCM

(Mackey et al. 2015) and, moreover, are able to exploit some of the host respiratory $CO_2$

as an alternative DIC source (Stambler 2011). Beneficial effects of high $pCO_2$ on

community carbon production, but not oxygen production, for shallow water coral reefs

have been reported by Langdon & Atkinson (2005), who found a 20–50% increase in

carbon production of coral assemblages composed of *Porites compressa* and *Montipora*



*capitata*. This result led to the hypothesis that increasing $CO_2$ causes a decrease in the

photosynthetic quotient of corals, which could be a product of the metabolism of the coral

host if it favors the production of carbohydrates over proteins and lipids (Langdon &

Atkinson 2005). While this hypothesis is appealing, it was not possible to test in the

present study because $P_{net}$ was determined through measurements of $O_2$ (see Material and

Methods).

       In our three experiments, maximal community $G_{net}$ was coincident with the

highest PAR. Similar to $P_{net}$, the relationships of $G_{net}$ with PAR at the two $pCO_2$ levels

were best-fit by a hyperbolic tangent function. The lack of changes in the parameters of

these relationships as a result of the treatment conditions demonstrated that $pCO_2$ and

light did not have interactive effects on $G_{net}$ (Table 2). Only the elevations of the

hyperbolic functions for the two habitats in Moorea were affected by high $pCO_2$, and in

this case their reduction relative to ambient $pCO_2$ demonstrates that $G_{net}$ consistently was

lower, regardless of PAR, at high $pCO_2$. Comparative data on the effect of PAR on the

response of community calcification to $pCO_2$ are currently not available, but the few

studies of these effects that have been conducted at the organism scale report equivocal

results (Marubini et al. 2001; Comeau et al. 2013; Dufault et al. 2013; Sugget et al. 2013;

Comeau et al. 2014b; Enochs et al. 2014).


       The consistently lower $G_{net}$ in the high $pCO_2$ treatments for the three experiments

could have resulted from either a decrease in gross calcification, an increase in

dissolution, or a combination of both. The constant offset (i.e., difference in elevation of

the response) between $G_{net}$ under ambient and high pCO$_2$ at any given PAR suggests the

effect cannot be accounted for solely by changes in gross calcification ($G_{gross}$). Indeed, if

only $G_{gross}$ was affected, a proportional effect $G_{net}$ would be expected, with the reduction

of $G_{net}$ associated with high pCO$_2$ varying with $G_{gross}$ and therefore PAR.  In contrast, if

dissolution and bioerosion, which are mostly chemically and mechanically driven

(Andersson and Gledhill 2013), were responsible for the reduced $G_{net}$ at high pCO$_2$, it is

likely that PAR would have only a small influence in $G_{net}$. Thus, it is likely that

increasing dissolution/bierosion in the high pCO$_2$ treatment caused most of the observed

decreases in $G_{net}$.

Although the two coral reef communities studied in Moorea differed in

substratum composition (i.e., sand in the back reef versus pavement in the outer reef),

community structure, and the quality and quantity of light applied (i.e., blue-biased at

depth and a 40% reduction at 17-m versus 2-m depth), both communities exhibited a 50-

60% decline in $G_{net}$ at 1300 μatm pCO$_2$. In contrast, mean $G_{net}$ for the Oahu back reef

community was less affected by pCO$_2$ than for the communities of Moorea. The reduced

sensitivity of $G_{net}$ to ~ 1200 μatm pCO$_2$ for back reef communities in Oahu may reflect

different sediment composition and legacy effects associated with environmental

conditions in the bay from which the organisms and sediment were collected. Critically,

the organisms for the Oahu experiment were collected from Kaneohe Bay where seawater

pCO$_2$ (up to ~450-500 μatm) is higher than current atmospheric levels (~400 μatm), and

there are strong diurnal cycles in pCO$_2$, and rapid changes in pCO$_2$ during storm events

(Fagan and Mackenzie 2007; Drupp et al. 2011). These conditions potentially could have

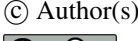



created the opportunity for physiological acclimatization that might reduce their sensitivity to high $pCO_2$ in the experimental trials.

The relationship between community $P_{net}$ and $G_{net}$ is used commonly to evaluate the "state" of a coral reef (Gattuso et al. 1999; Lantz et al. 2014), with coral reefs dominated by high coral cover and low cover of macroalgae characterized by elevated slopes of the $P_{net}$ - $G_{net}$ relationship. In the present study, the slopes of the relationships between $P_{net}$ and $G_{net}$ in the ambient treatment were between 0.14 (Oahu) (this and all

following slope values have units of mmol $CaCO_3$ mmol $O_2^{-1}$) and 0.27 (Moorea fore reef). In Moorea, the slopes were higher for the fore reef (0.27 and 0.30) versus the back reef (0.17 and 0.18) community, which demonstrated that $G_{net}$ was more sensitive to changes in $P_{net}$ in fore reef communities, probably because of a higher calcifier cover. The slopes of the $P_{net}$ - $G_{net}$ relationships for the communities tested are within the range

estimated from in situ "reef scale" measurements, which indicate a mean value of 0.22 based on 52 reefs (Gattuso et al. 1999). More recently, Shaw et al. (2012) reported a $P_{net}$ - $G_{net}$ slope of 0.24 for the reef flat of Lady Elliot Island, Australia, and a slope of 0.14 was reported for Ningaloo reef, Australia (Falter et al. 2012). The consistency between the slopes reported herein and values determined in situ (e.g., Shaw et al. 2012, Gattuso et al.

1999) suggest that our constructed communities, and the conditions to which they were exposed, reproduced conditions found in situ on coral reefs. This outcome lends support to the inferences we are able to make regarding the response of reef communities to elevated $pCO_2$ for which currently there is no in situ data.



Our results are consistent with the hypothesis that OA will affect the relationship

between community $P_{net}$ and $G_{net}$ (sensu Gattuso et al. 1999) because elevations of the

$P_{net}$ - $G_{net}$ relationships varied between treatments and were greater under ambient $pCO_2$.

The absence of changes in slopes as a function of $pCO_2$ probably was due to the lack of a

$pCO_2$ effect on $P_{net}$, and the lack of a PAR-$pCO_2$ interactive effect on $P_{net}$ and $G_{net}$.

Furthermore, the community composition remained the same in the ambient and elevated

$pCO_2$ conditions, with no mortality or loss of benthic cover of living organisms during

the course of the experiment, which could potentially have modified the community $P_{net}$ -

$G_{net}$ relationship (Lantz et al. 2014; Shaw et al. 2015) due to taxon-specific $P_{net}$ - $G_{net}$

relationships (C.A. Lantz unpubl.). Thus, this result indicates that elevated $CO_2$ alone

(e.g., without considering warming) can modify the balance between calcification and

photosynthesis at the scale of a whole reef, because of a decrease in coral reef community

calcification while photosynthesis remains constant.



**Acknowledgements**

We thank RD Gates and HM Putnam for access to the infrastructure of HIMB and

laboratory assistance in Hawaii. This study was funded by the National Science

Foundation (OCE 10-41270 and 14-15268) and the Moorea Coral Reef LTER (OCE 04-

17413 and 10-26852). This is contribution number 235 of the CSUN Marine Biology

Program. S. Comeau was supported by ARC (Discovery Early Career Researcher Award;

DE160100668) during the writing of the manuscript.

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

**Table 1:** Mean carbonate chemistry and temperature treatments in the flumes during the experiments conducted with back reef communities in Moorea and Oahu, and the fore reef community in Moorea. The mean $\pm$ SE partial pressure of $CO_2$ (p$CO_2$), and the saturation states of aragonite ($\Omega_{arag}$) were calculated from $pH_T$, total alkalinity ($A_T$), salinity (S) and temperature (T). SE for salinity was < 0.1.

| Experiment | Treatment | $pH_T$ | $A_T$ | p$CO_2$ | $C_T$ | $\Omega_{arag}$ | S | T |
|---|---|---|---|---|---|---|---|---|
| **Moorea Back reef** | **Ambient** | 8.01 $\pm$ 0.02 | 2339 $\pm$ 2 | 453 $\pm$ 30 | 2025 $\pm$ 9 | 3.52 $\pm$ 0.09 | 35.9 | 26.9 $\pm$ 0.1 |
| | **OA** | 7.61 $\pm$ 0.01 | 2344 $\pm$ 1 | 1317 $\pm$ 50 | 2230 $\pm$ 7 | 1.64 $\pm$ 0.06 | 35.9 | 27.0 $\pm$ 0.1 |
| **Oahu Back reef** | **Ambient** | 7.96 $\pm$ 0.01 | 2160 $\pm$ 4 | 490 $\pm$ 23 | 1936 $\pm$ 8 | 2.59 $\pm$ 0.06 | 33.4 | 23.9 $\pm$ 0.2 |
| | **OA** | 7.62 $\pm$ 0.02 | 2164 $\pm$ 4 | 1233 $\pm$ 76 | 2074 $\pm$ 12 | 1.36 $\pm$ 0.10 | 33.4 | 23.9 $\pm$ 0.2 |
| **Moorea Fore reef** | **Ambient** | 8.04 $\pm$ 0.01 | 2329 $\pm$ 2 | 400 $\pm$ 14 | 1992 $\pm$ 8 | 3.71 $\pm$ 0.08 | 36.5 | 27.1 $\pm$ 0.1 |
| | **OA** | 7.65 $\pm$ 0.01 | 2330 $\pm$ 2 | 1176 $\pm$ 37 | 2198 $\pm$ 6 | 1.75 $\pm$ 0.05 | 36.5 | 27.0 $\pm$ 0.1 |





**Table 2:** Results of the t-tests used to compare between $pCO_2$ treatments the parameters

of the hyperbolic tangent functions describing the relationship between community net

photosynthesis ($P_{net}$) and PAR and net calcification ($G_{net}$) and PAR. Parameters of the

hyperbolic function are the maximum rate *($P_{net\ max}$ and $G_{net\ max}$)*, the slope of the initial

portion of the relationship ($\alpha$), and the intercept ($C_0$).


| Parameter | Experiment | Function parameter | *p*-value |
|---|---|---|---|
| Net Photosynthesis ($P_{net}$) | Moorea – Back reef | $P_{net\ max}$ | 0.558 |
| | | $\alpha$ | 0.387 |
| | | $C_0$ | 0.559 |
| | Oahu – Back reef | $P_{net\ max}$ | 0.840 |
| | | $\alpha$ | 0.536 |
| | | $C_0$ | 0.621 |
| | Moorea – Fore reef | $P_{net\ max}$ | 0.942 |
| | | $\alpha$ | 0.792 |
| | | $C_0$ | 0.579 |
| Net Calcification ($G_{net}$) | Moorea – Back reef | $G_{net\ max}$ | 0.376 |
| | | $\alpha$ | 0.836 |
| | | $C_0$ | 0.046 |
| | Oahu – Back reef | $P_{net\ max}$ | 0.867 |
| | | $\alpha$ | 0.126 |
| | | $C_0$ | 0.394 |
| | Moorea – Fore reef | $P_{net\ max}$ | 0.736 |
| | | $\alpha$ | 0.715 |
| | | $C_0$ | 0.002 |





**Table 3:** Results of ANCOVA analyses testing for effects of $pCO_2$ on slopes and elevation of the $P_{net}$ - $G_{net}$ relationships. Results are shown for the experiments with back reef communities in Moorea and Oahu, and the fore reef communities in Moorea.

| Experiment | Parameter | SS | F-value | P-value |
|---|---|---|---|---|
| Moorea - back reef | Slope | 0.50 | $F_{1,92} = 0.10$ | 0.749 |
| | Elevation | 100.48 | $F_{1,92} = 20.49$ | <0.001 |
| Oahu - back reef | Slope | 6.10 | $F_{1,74} = 1.15$ | 0.286 |
| | Elevation | 66.40 | $F_{1,74} = 12.57$ | <0.001 |
| Moorea - fore reef | Slope | 0.83 | $F_{1,62} = 0.24$ | 0.623 |
| | Elevation | 36.39 | $F_{1,62} = 10.61$ | 0.002 |


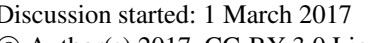



**Fig. 1.** Relationships of net primary production ($P_{net}$) with PAR in three coral reef

communities representing the back reef of Moorea (A), the back reef of Oahu (B), and

the fore reef of Moorea (C). Communities were incubated under ambient $pCO_2$ (~400

µatm, black symbols and lines) and elevated $pCO_2$ (~1200 µatm, red symbols and lines).

The curves represent the best fit of a hyperbolic tangent function for the relationship

between $P_{net}$ with PAR.

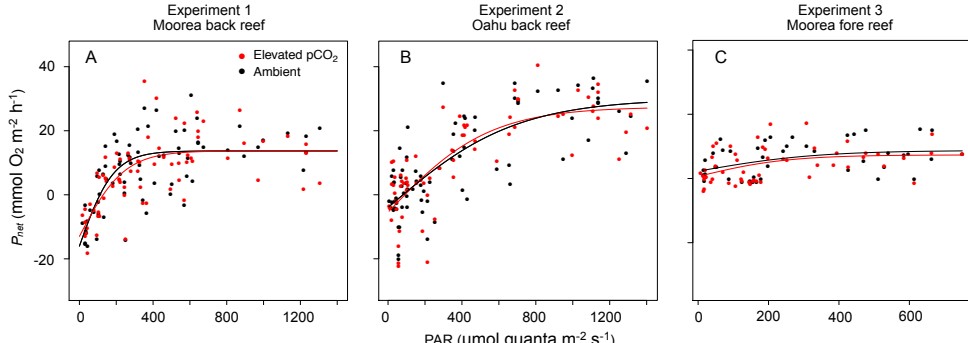


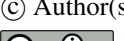


**Fig. 2.** Relationships of net calcification ($G_{net}$) with PAR in three coral reef communities

representing the back reef of Moorea (A), the back reef of Oahu (B), and the fore reef of

Moorea (C). Communities were incubated under ambient $pCO_2$ (~400 μatm, black

symbols and lines) and elevated $pCO_2$ (~1200 μatm, red symbols and lines). The curves

represent the best fit of a hyperbolic tangent function for the relationship between $G_{net}$

and PAR.

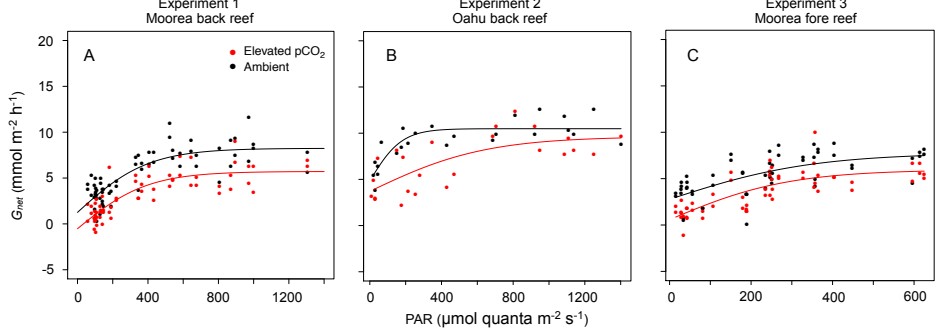



**Fig. 3.** Variations in $G_{net}$ as a function of $P_{net}$ in the three study sites: (A) Moorea back

reef, (B) Oahu back reef, and (C) Moorea fore reef. Relationships were determined under

control pCO$_2$ (400 µatm, black points and lines) and elevated pCO$_2$ (~1200 µatm, red

points and lines). For the three communities and the two pCO$_2$ levels the slopes of the

linear relationships between $P_{net}$ and $G_{net}$ were significant.

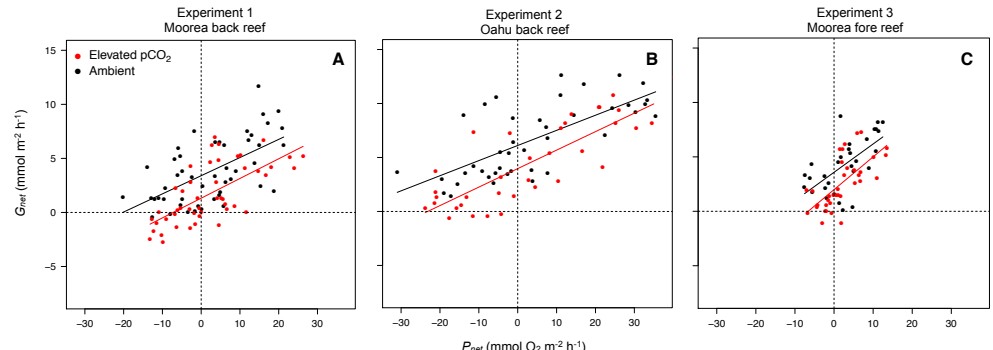
