# Peer review of "Daily variation in net primary production and net calcification in coral reef communities exposed to elevated $pCO_2$"

_Biogeosciences, 2017_

## Referee Comment (RC1) · Anonymous Referee #1 · 22 Mar 2017

General Comments:

Comeau et al. present results from three flume studies, a backreef from Moorea, a backreef from Oahu, and a forereef from Moorea. They present statistical models relating net production and net calcification to PAR for ambient and high CO2 conditions. They then analyze the relationship between production and calcification at ambient and high CO2 conditions. They conclude that ocean acidification conditions depress calcification without changing production.

The paper is well-written with logical sequences and transitions. I think the paper is structurally sound and well-organized. The figures and tables are clean and easy-to-read. However, I have a difficult time discerning whether the data and information

presented in this study is new data worthy of a separate publication. As the authors acknowledge, the data for two out of the three flume studies reported in this manuscript were already discussed and presented in earlier manuscripts (Comeau et al. 2015 Biogeosciences-Moorea backreef, Comeau et al. 2016a Global Change Biology-Moorea forereef). Each of these two previously published studies present daytime, nighttime, and daily integrated calcification results for the mixed community discussed in this study, macrocalcifiers (i.e. corals + CCA), and sediment/pavement alone. Although this study presents higher resolution calcification data and new production data for the Moorea sites, plus all of the data for the Oahu site, the analysis is still limited and does little to extend beyond the two previously published studies. Furthermore, the conclusions from this paper, that ocean acidification depresses net calcification independently of production, suggests that enhanced dissolution is largely responsible for driving the decreased net calcification at high CO2 treatment levels. This conclusion can be inferred the data and figures in Comeau et al. 2015 and Comeau et al. 2016a.

Overall, I don't think there are major problems with the technical quality of the manuscript. I just don't know whether there is enough original material presented here to stand alone as a manuscript. I will leave that for the editor and co-reviewers to decide and for the authors to better justify.

Specific Comments:

Methods:

The section about flume design and setup seems very long and repetitive from previously published papers (lines 140-205). I recommend that you cite the original references as necessary and replace much of the text with a diagram showing all the important features (sites, numbers of flume replicates, flume lengths, flow speeds, CO2 treatments, pictures of community composition in the flumes, etc.). I think it will be much easier and faster for the reader to absorb the material this way.

Please include information about oxygen sensor calibration and accuracy determination (similar to what you've done for TA and pH measurements).

Please provide more details to justify your omission of air/sea gas fluxes.

Results:

Have you tested whether your hyperbolic tangent model for PAR-production and for PAR-calcification provides significantly more explanatory power than a null model? Experiments 1 and 2 for the PAR-production model (Fig. 1a and 1b) seem to fit the expected behavior of a hyperbolic tangent (or Michaelis-Menten) model, but Experiment 3 (the deep Moorea forereef) does not (Fig. 1c). Using AIC will help you identify the most parsimonious model type, but not tell you whether the model significantly increases the explanatory power of the data. In the case that any of the models for any of the $CO_2$ treatments do not provide additional explanatory power over the null model, you are going to have to think about why not. In general, I think you should be providing the results of all model parameter fits (including mean estimates and standard errors) as a supplemental table for interested readers. Providing only your t-test results between ambient and high $CO_2$ treatments does not provide the reader with information about the actual parameter values nor allow the reader to assess whether these models are even appropriate fits to the data in the first place (i.e. whether the model parameters are significantly different from zero).

Why did you choose an ANCOVA approach for analyzing the calcification-production relationship at ambient and high $CO_2$ as opposed to simple linear regressions, where you could look for changes in slope and intercept in ambient vs. high $CO_2$ conditions? I think linear regressions are more intuitive, but perhaps there is a logic behind the ANCOVA approach that just needs to be explained further? The presentation of the ANCOVA results (Section 3.2) could largely be collapsed into a table that gives slope and intercept mean estimates and standard errors for all experiments and treatments (as opposed to Table 3 which currently just lists results of tests for differences in parameter estimates between ambient and high $CO_2$ treatments for the three experiments).

Discussion:

I recommend that you re-structure the Discussion section to focus on your positive re-
sult (depressed net calcification at high CO2 and its possible mechanisms) first before
discussing your null result (no changes in production at high CO2 or in PAR-production
relationship).

---

## Referee Comment (RC2) · Anonymous Referee #2 · 30 Mar 2017

Comeau et al. conduct flume experiments to examine the effects of elevated pCO2 on reef community calcification, photosynthesis, respiration, and the parameters of the mathematical fits of these response variables under varied irradiance, as well as the relationships among Pnet, Gnet, and PAR. While portions of the data have been reported previously, the data from the experiments on Oʻahu are new, as are many of the analyses and Pnet data. Overall I think this is a nice paper which deserves to be published. My comments are largely for minor revisions, as detailed below.

First, Tahitian and Hawaiian names are consistently misspelled throughout the manuscript and should be corrected:

Hawaii should be Hawaiʻi Oahu should be Oʻahu Kaneohe should be KĀĄneʻohe

(hmmm, the conversion software seems to be rendering Kane'ohe in a strange way here...the "a" should have a kahako, or macron, above it, indicating stress) Moorea should be Mo'orea

Line 225, I believe that the authors may have convoluted CO2-eq (CO2 equivalents, which includes the radiative forcing provided by CO2, CH4, N2O, SO2, etc. in common currency that is proportional to the radiative forcing provided by an equivalent concentration of CO2) under RCP 8.5 with atmospheric CO2 concentration. Under RCP 8.5, the projection for the end of this century is for CO2-eq to reach nearly 1300 uatm (e.g., Moss et al., 2010), but only a portion of that forcing is provided by CO2. Under RCP 8.5, the projection is for atmospheric CO2 to reach about 930 uatm at the end of this century, and to reach 1300 uatm around the middle of next century. See, for example:

Meinshausen et al. 2011. The RCP greenhouse gas concentrations and their extensions from 1765 to 2300. Climatic Change.

Line 230, It's commendable that the authors allowed the pH to vary somewhat over the diel cycle, similar to what happens in nature, but it's also important to consider that acidification results in *non-linear* changes in pH. This non-linearity is related to reduced seawater buffer capacity under acidification. OA results in an offset in seawater DIC, and this offset translates into variable changes in pH depending on the starting conditions. Given the daytime treatment chemistries listed in Table 1 and a 0.1 pH reduction at night under present-day conditions (similar to conditions in the field), I used co2sys to estimate that the corresponding pH reduction in the acidified treatments (in order to maintain the equivalent DIC offset) would be about 0.15 at night. See:

Jury et al. 2013. Buffer capacity, ecosystem feedbacks, and seawater chemistry under global change. Water.

Shaw et al. 2013. Anthropogenic changes to seawater buffer capacity combined with natural reef metabolism induce extreme future coral reef CO2 conditions. Global Change Biology.

Line 255, Since pH was kept stable by the controllers while TA was varying due to calcification/carbonate dissolution, DIC was not constant during the incubations. Perhaps provide some estimate of how much DIC and TA varied during the course of the incubations?

Line 345, Please report the AIC values that justify the use of hyperbolic tangent function over log or linear functions. The hyperbolic tangent fit seems sensible for experiments 1 and 2, but for experiment 3 the relationship looks much closer to linear, and surprisingly flat actually. Also, I would suggest using AICc (corrected AIC) as it reduces the chances of overfitting with finite sample sizes. Since AICc converges to AIC with very large sample size anyway, AICc is probably best used as the default between the two.

Is Fig. 1 showing the Pnet values from the daytime only (and none from the night)? If yes, why aren't the nighttime data included as well? Either way, please clarify and add this information to the figure legend. You'd expect that Pnet should be negative at night (net respiration) under zero irradiance and it should eventually rise to Pnet=0 at some low level of irradiance (the compensation point). That pattern pops out in experiment 1 and to a lesser extent in experiment 2, but it looks like in experiment 3 the fits show positive Pnet even under zero irradiance. At a minimum, that seems strange. . .

Line 350, Ditto the above on AIC, AICc, justifying the model selection, daytime vs. nighttime data, and modifying the legend for Fig. 2 to clarify.

Line 380, I believe "back reef communities" should be "fore reef community", correct?

Line 448, Should be, "an increase in [carbonate] dissolution".

Line 453, Carbonate dissolution would be detectable by changes in chemistry measured, but mechanical bioerosion would not be.

Line 456, Likewise, changes in mechanical bioerosion can't be detected via chemical methods.

Line 472, Physiological acclimatization to high pCO2 in KĀA̧neʻohe Bay is a possible

explanation for reduced sensitivity to acidification, but so is local adaptation.

---

## Referee Comment (RC3) · Anonymous Referee #2 · 26 Apr 2017

p.s. The more I think about it, the more concerned I am that we, as a field, might be missing some really important changes if we are measuring Pnet via O2 flux only or DIC flux only. The photosynthetic quotient does appear to be quite (potentially) labile for reef organisms, and probably for many others. In this study, Pnet can only be estimated via O2 flux, not DIC flux, but as work by Langdon and Atkinson (2005) shows, examining one paramter or the other can yield radically different conclusions about how Pnet may or may not change under perturbation. The authors acknowledge and discuss this issue, but I would encourage them to emphasize the issue to a much greater extent–this question probably should dictate how the field measures Pnet going forward (both O2 flux and DIC flux, ideally), and the authors have an excellent opportunity to

make that case here.

---

## Author Comment (AC1) · 2 Jun 2017

**Anonymous Referee 1 General Comments:** Comeau et al. present results from three flume studies, a backreef from Moorea, a backreef from Oahu, and a forereef from Moorea. They present statistical models relating net production and net calcification to PAR for ambient and high CO2 conditions. They then analyze the relationship between production and calcification at ambient and high CO2 conditions. They conclude that ocean acidification conditions depress calcification without changing production. The paper is well-written with logical sequences and transitions. I think the paper is structurally sound and well-organized. The figures and tables are clean and

easy-to- read. However, I have a difficult time discerning whether the data and information presented in this study is new data worthy of a separate publication. As the authors acknowledge, the data for two out of the three flume studies reported in this manuscript were already discussed and presented in earlier manuscripts (Comeau et al. 2015 Biogeosciences-Moorea backreef, Comeau et al. 2016a Global Change Biology-Moorea forereef). Each of these two previously published studies present daytime, nighttime, and daily integrated calcification results for the mixed community discussed in this study, macrocalcifiers (i.e. corals + CCA), and sediment/pavement alone. Although this study presents higher resolution calcification data and new production data for the Moorea sites, plus all of the data for the Oahu site, the analysis is still limited and does little to extend beyond the two previously published studies. Furthermore, the conclusions from this paper, that ocean acidification depresses net calcification independently of production, suggests that enhanced dissolution is largely responsible for driving the decreased net calcification at high CO2 treatment levels. This conclusion can be inferred the data and figures in Comeau et al. 2015 and Comeau et al. 2016a. Overall, I don't think there are major problems with the technical quality of the manuscript. I just don't know whether there is enough original material presented here to stand alone as a manuscript. I will leave that for the editor and co-reviewers to decide and for the authors to better justify.

**Response to the General comment:** We thank Reviewer1 for her/his comments and for noting that our paper is well written, structurally sound, and well organized. As pointed out by reviewer1 the data presented in this manuscript includes calcification data for Moorea that were previously published. Nevertheless we believe that our paper, which includes data from 3 large-scale experiments over two distinct geographical locations, is novel for several reasons. First, as noted by reviewer1, all the data for Oahu and all the Pnet data (Oahu, Moorea back reef, and Moorea fore reef) have not been described elsewhere. As a result 66 percent of the presented data are novel. Second, our manuscript focuses on the response of community Pnet to OA for which data are critically needed. Third, only mean calcification data for Moorea were presented in

prior publications, whereas here we discuss the relationships between irradiance and calcification as well as the relationship primary production and calcification for both Moorea and Hawaii. Therefore, our paper combines unpublished data and a subset of published data to provide a comparison of the response of Gnet and Pnet of three distinctive communities exposed to similar pCO2 levels. Finally, we feel it is important to note that reviewer2 believes that "while portions of the data have been reported previously, the data from the experiments on Oʻahu are new, as are many of the analyses and Pnet data." and therefore does not share the novelty concerns of reviewer1 and recommend our paper for publication.

**Specific Comments: Methods: Comment 1:** The section about flume design and setup seems very long and repetitive from previously published papers (lines 140-205). I recommend that you cite the original references as necessary and replace much of the text with a diagram showing all the im- portant features (sites, numbers of flume replicates, flume lengths, flow speeds, CO2 treatments, pictures of community composition in the flumes, etc.). I think it will be much easier and faster for the reader to absorb the material this way.

**Response 1:** Good suggestion. The methods have been shortened and a diagram added (Figure 1).

**Comment 2:** Please include information about oxygen sensor calibration and accuracy determination (similar to what you've done for TA and pH measurements). Please provide more details to justify your omission of air/sea gas fluxes.

**Response 2:** As suggested more details on the oxygen sensors are now provided in the materials and methods.

**Results: Comment 3:** Have you tested whether your hyperbolic tangent model for PAR- production and for PAR-calcification provides significantly more explanatory power than a null model? Experiments 1 and 2 for the PAR-production model (Fig. 1a and 1b) seem to fit the expected behavior of a hyperbolic tangent (or Michaelis-

Menten) model, but Experiment 3 (the deep Moorea forereef) does not (Fig. 1c). Using AIC will help you identify the most parsimonious model type, but not tell you whether the model significantly increases the explanatory power of the data. In the case that any of the models for any of the CO2 treatments do not provide additional explanatory power over the null model, you are going to have to think about why not. In general, I think you should be providing the results of all model parameter fits (including mean estimates and standard errors) as a supplemental table for interested readers. Providing only your t- test results between ambient and high CO2 treatments does not provide the reader with information about the actual parameter values nor allow the reader to assess whether these models are even appropriate fits to the data in the first place (i.e. whether the model parameters are significantly different from zero).

**Response 3.** There was an effect of PAR on production and calcification on the fore reef community and any of the tested models (linear, logarithmic and hyperbolic tangent) provided more explanatory power than a null model. Because the three models had similar explanatory powers to model the relationship between PAR and Pnet/Gnet on the fore reef, we decided to use a hyperbolic tangent model to facilitate comparisons with the experiment 1 and 2 for which this type of model was best suited.

**Comment 4.** Why did you choose an ANCOVA approach for analyzing the calcification-production relationship at ambient and high CO2 as opposed to simple linear regressions, where you could look for changes in slope and intercept in ambient vs. high CO2 conditions? I think linear regressions are more intuitive, but perhaps there is a logic behind the ANCOVA approach that just needs to be explained further? The presentation of the ANCOVA results (Section 3.2) could largely be collapsed into a table that gives slope and intercept mean estimates and standard errors for all experiments and treatments (as opposed to Table 3 which currently just lists results of tests for differences in param- eter estimates between ambient and high CO2 treatments for the three experiments).

**Response 4.** Results of the ANCOVA presented in Table 3 actually corresponds to

what Reviewer1 is asking for as the ANCOVA test for differences in slopes and elevation (intercept) at ambient and elevated pCO2. For clarification, we took into account the suggestion of reviewer1 and Table 3 has been changed to show the mean estimates and standard errors for all experiments and treatments. Results from the ANCOVA are now only presented in the text.

**Discussion: Comment 5:** I recommend that you re-structure the Discussion section to focus on your positive result (depressed net calcification at high CO2 and its possible mechanisms) first before discussing your null result (no changes in production at high CO2 or in PAR- production relationship).

**Response 5.** While the positive results are of high importance, we believe that the main novelty of our paper is to show that there is no change in Pnet. Therefore we respectfully prefer to keep the discussion initial structure.

**Anonymous Referee 2 General Comments:** Comeau et al. conduct flume experiments to examine the effects of elevated pCO2 on reef community calcification, photosynthesis, respiration, and the parameters of the mathematical fits of these response variables under varied irradiance, as well as the relationships among Pnet, Gnet, and PAR. While portions of the data have been reported previously, the data from the experiments on Oʻahu are new, as are many of the analyses and Pnet data. Overall I think this is a nice paper which deserves to be published. My comments are largely for minor revisions, as detailed below.

**Response to the General Comment from reviewer 2:** We thank reviewer2 for his/her comments and we are please to note that reviewer2 recommends our paper for publication in Biogeosciences.

**Comment 1:** First, Tahitian and Hawaiian names are consistently misspelled throughout the manuscript and should be corrected: Hawaii should be Hawaiʻi Oahu should be Oʻahu Kaneohe should be KAĪLA Ìĭneʻohe (hmmm, the conversion software seems to be rendering Kaneʻohe in a strange way here...the "a" should have a kahako, or

macron, above it, indicating stress) Moorea should be Moʻorea

**Response 1:** As suggested this has been corrected in the manuscript.

**Comment 2:** Line 225, I believe that the authors may have convoluted CO2-eq (CO2 equivalents, which includes the radiative forcing provided by CO2, CH4, N2O, SO2, etc. in common currency that is proportional to the radiative forcing provided by an equivalent concen- tration of CO2) under RCP 8.5 with atmospheric CO2 concentration. Under RCP 8.5, the projection for the end of this century is for CO2-eq to reach nearly 1300 uatm (e.g., Moss et al., 2010), but only a portion of that forcing is provided by CO2. Under RCP 8.5, the projection is for atmospheric CO2 to reach about 930 uatm at the end of this century, and to reach 1300 uatm around the middle of next century. See, for example: Meinshausen et al. 2011. The RCP greenhouse gas concentrations and their extensions from 1765 to 2300. Climatic Change.

**Response 2:** Thank you for this comment; the revised version of the manuscript has been changed accordingly.

**Comment 3:** Line 230, It's commendable that the authors allowed the pH to vary somewhat over the diel cycle, similar to what happens in nature, but it's also important to consider that acidification results in *non-linear* changes in pH. This non-linearity is related to reduced seawater buffer capacity under acidification. OA results in an offset in seawater DIC, and this offset translates into variable changes in pH depending on the starting conditions. Given the daytime treatment chemistries listed in Table 1 and a 0.1 pH reduction at night under present-day conditions (similar to conditions in the field), I used co2sys to estimate that the corresponding pH reduction in the acidified treatments (in order to maintain the equivalent DIC offset) would be about 0.15 at night. See: Jury et al. 2013. Buffer capacity, ecosystem feedbacks, and seawater chemistry under global change. Water. Shaw et al. 2013. Anthropogenic changes to seawater buffer capacity combined with natural reef metabolism induce extreme future coral reef CO2 conditions. Global Change Biology.

**Response 3:** In our experiment we decided to apply the same offset under present and future conditions in order to limit the number of confounding effects. We do agree that, in the future ocean, fluctuations in pH will be larger (if the ratio of autotroph to heterotroph remains unchanged) due to changes in the buffering capacity of seawater. We have actually tested previously the response of organisms to these expected larger variations in pH (Comeau et al. 2014, MEPS). Larger pH variations would likely even further enhance the response observed in our study. This is now discussed in the manuscript.

**Comment 4:** Line 255, Since pH was kept stable by the controllers while TA was varying due to calcification/carbonate dissolution, DIC was not constant during the incubations. Per- haps provide some estimate of how much DIC and TA varied during the course of the incubations?

**Response 4:** The duration of the incubations was chosen to limit the changes in TA by only < 5 percent ( 40-50 $\mu$mol kg-1). As a result changes in DIC were also limited to < 5 percent, which likely did not impact the organisms. This is now included in the manuscript.

**Comment 5:** Line 345, Please report the AIC values that justify the use of hyperbolic tangent function over log or linear functions. The hyperbolic tangent fit seems sensible for experiments 1 and 2, but for experiment 3 the relationship looks much closer to linear, and surprisingly flat actually. Also, I would suggest using AICc (corrected AIC) as it reduces the chances of overfitting with finite sample sizes. Since AICc converges to AIC with very large sample size anyway, AICc is probably best used as the default between the two.

**Response 5:** Actually we already used an AICc and not an AIC for our study. It has been corrected in the revised version of the manuscript.

**Comment 6:** Is Fig. 1 showing the Pnet values from the daytime only (and none from the night)? If yes, why aren't the nighttime data included as well? Either way, please

clarify and add this information to the figure legend. You'd expect that Pnet should be negative at night (net respiration) under zero irradiance and it should eventually rise to Pnet=0 at some low level of irradiance (the compensation point). That pattern pops out in experiment 1 and to a lesser extent in experiment 2, but it looks like in experiment 3 the fits show positive Pnet even under zero irradiance. At a minimum, that seems strange.

**Response 6:** Yes, Fig 1 shows only the Pnet values measured when PAR was > 0. This was done because the goal was to explore the relationship between PAR and Pnet. If night data were included, half of the points would have been measured when PAR= 0 which would have artificially biased our result. Actually the first measurements of Pnet shown on the plots are not done at PAR = 0, but PAR > 8 umol quanta m-2s-1. Therefore this demonstrates that fore reef communities which are acclimated to lower PAR and are capable of positive Pnet at very low PAR. This is now discussed in the manuscript.

**Comment 7:** Line 350, Ditto the above on AIC, AICc, justifying the model selection, daytime vs. nighttime data, and modifying the legend for Fig. 2 to clarify.

**Response 7:** Done

**Comment 8:** Line 380, I believe "back reef communities" should be "fore reef community", correct?

**Response 8:** No, "back reef communities" is correct.

**Comment 9:** Line 448, Should be, "an increase in [carbonate] dissolution".

**Response 9:** Corrected.

**Comment 10:** Line 453, Carbonate dissolution would be detectable by changes in chemistry measured, but mechanical bioerosion would not be.

**Comment 11:** Line 456, Likewise, changes in mechanical bioerosion can't be detected

via chemical methods.

**Response 10 and 11:** This is correct, it is now included in the discussion.

**Comment 12:** Line 472, Physiological acclimatization to high pCO2 in KAÌLA ÌÍne'ohe Bay is a possible explanation for reduced sensitivity to acidification, but so is local adaptation.

**Response 12:** Good point, this is now included.

**Additional comment :** The more I think about it, the more concerned I am that we, as a field, might be missing some really important changes if we are measuring Pnet via O2 flux only or DIC flux only. The photosynthetic quotient does appear to be quite (potentially) labile for reef organisms, and probably for many others. In this study, Pnet can only be estimated via O2 flux, not DIC flux, but as work by Langdon and Atkinson (2005) shows, examining one parameter or the other can yield radically different conclusions about how Pnet may or may not change under perturbation. The authors acknowledge and discuss this issue, but I would encourage them to emphasize the issue to a much greater extent–this question probably should dictate how the field measures Pnet going forward (both O2 flux and DIC flux, ideally), and the authors have an excellent opportunity to make that case here.

**Response to the Additional comment:** We fully agree with Reviewer2 comment. The apparent contradiction between studies looking at the effects of OA on coral calcifiers and coral communities Pnet could be partly due to the use of DIC vs O2 fluxes. This point is indeed critical and should be further studied in the future at the organismal and community level. As suggested this is now further discussed in our manuscript.